# Darknet Traffic Big-Data Analysis and Network Management for Real-Time Automating of the Malicious Intent Detection Process by a Weight Agnostic Neural Networks Framework

**Konstantinos Demertzis [1,2,\*]**, **Konstantinos Tsiknas [3]**, **Dimitrios Takezis [4]**, **Charalabos Skianis [5]** and **Lazaros Iliadis [2]**

1    Laboratory of Complex Systems, Department of Physics, Faculty of Sciences, Kavala Campus, International Hellenic University, 65404 St. Loukas, Greece
2    Department of Civil Engineering, School of Engineering, Faculty of Mathematics Programming and General Courses, Democritus University of Thrace, Kimmeria, 67100 Xanthi, Greece; liliadis@civil.duth.gr
3    Department of Electrical and Computer Engineering, Democritus University of Thrace, 67100 Xanthi, Greece; ktsiknas@ee.duth.gr
4    Hellenic National Defence General Staff, Stratopedo Papagou, Mesogeion 227-231, 15561 Athens, Greece; d.taketzis@hndgs.mil.gr
5    Department of Information and Communication Systems Engineering, University of Aegean, 83200 Karlovassi, Greece; cskianis@aegean.gr
*    Correspondence: kdemertzis@teiemt.gr

**Abstract:** Attackers are perpetually modifying their tactics to avoid detection and frequently leverage legitimate credentials with trusted tools already deployed in a network environment, making it difficult for organizations to proactively identify critical security risks. Network traffic analysis products have emerged in response to attackers' relentless innovation, offering organizations a realistic path forward for combatting creative attackers. Additionally, thanks to the widespread adoption of cloud computing, Device Operators (DevOps) processes, and the Internet of Things (IoT), maintaining effective network visibility has become a highly complex and overwhelming process. What makes network traffic analysis technology particularly meaningful is its ability to combine its core capabilities to deliver malicious intent detection. In this paper, we propose a novel darknet traffic analysis and network management framework to real-time automating the malicious intent detection process, using a weight agnostic neural networks architecture. It is an effective and accurate computational intelligent forensics tool for network traffic analysis, the demystification of malware traffic, and encrypted traffic identification in real time. Based on a weight agnostic neural networks (WANNs) methodology, we propose an automated searching neural net architecture strategy that can perform various tasks such as identifying zero-day attacks. By automating the malicious intent detection process from the darknet, the advanced proposed solution is reducing the skills and effort barrier that prevents many organizations from effectively protecting their most critical assets.

**Keywords:** darknet; traffic analysis; network management; malicious intent detection; weight agnostic neural networks; real-time forensics; shapley value; power predicting score

## 1. Introduction

Interconnected heterogeneous information systems [1] exchange huge amounts of data per unit of time. This information consists of data at rest and data in motion. In the continuous flow model, the data arrive in successive streams in a continuous manner, resulting in it not being accessible by the storage mediums, either temporarily or permanently. Flow data are usually large in size, difficult to be processed in real-time, and when processed, they are either destroyed or archived and are very difficult to be recovered again, because the system's memory is typically very small.

The analysis, monitoring, and categorization of the Internet network traffic [2] is one of the most important tasks, and is characterized as a specialized solution and a valuable tool

that can be used not only to effectively deal with the design, management, and monitoring of the critical infrastructure of the system but also for the monitoring of attacks and the study of cybercrime [3].

The information exchanged can be requests, responses, or control data, fragmented in the form of network packets. When looking at individual network packets, it is extremely difficult to draw conclusions and exclude safe conclusions, because the information transmitted between devices on the network is fragmented into a number of packets, which are interconnected, containing all the information. This arbitrary and occasional nature of the collection of network traffic, while providing some information for drawing statistical conclusions, makes the use of typical mathematical analysis methods a rather difficult task that favors the network traffic modeling approach [4].

Many organizations, in their efforts to improve and enhance their security, collect as much web traffic data as possible, analyze it, by correlating it with the services they represent, and compare it with historical log files in order to optimize their decision-making process. By analyzing network traffic, safe conclusions can be drawn about the network, the users, and the total data usage, making it possible to model traffic in order to optimize network resources according to the monitoring needs and the control for legal and security issues [5,6]. More specifically, in cybersecurity, traffic analysis can be applied to secure services, guarantee critical data delivery, identify random sources of problems, adapt and optimize intrusion prevention and detection solutions, identify cybercriminals, and validate forensic data [7]. The major weaknesses associated with traffic packet analysis technologies are the following [8]:

1.  While the techniques are very effective, especially the Deep Packet Inspection (DPI) method in preventing Denial-Of-Service (DoS)/Distributed DoS (DDoS) attacks, buffer overflow attacks, and specific types of malware, they can also be used to create similar attacks from the adversary side, depending on their mode of operation;
2.  They add complexity to the operation of active network security methods and make them extremely difficult to manage. In addition, they increase the requirements for computing resources and introduce significant delays in online transactions, especially in encrypted traffic, because the latter requires the reconstruction of messages and entities at higher levels;
3.  Although there are many possible uses, an adverse situation is related to the ease with which someone can identify the recipient, or the sender of the content they are analyzing, raising privacy concerns.

They do not offer protection against zero-day attacks. The ever-increasing need for an organization to manage security incidents requires specialized analysis services, in order to fully understand the network environment and potential threats. This information, combined with cyber threat intelligence from the global threat landscape, allows for an informed and targeted response to cyber-related incidents [9].

In essence, the information ecosystem and the importance of its applications require the creation of a cybersecurity environment with fully automated solutions. These solutions include real-time incident handling, analysis, and other security information to identify known and unknown threats and reduce the risk for the critical data through a scalable troubleshooting and logging approach [10,11].

In this paper, we propose a novel darknet traffic analysis and network management framework for real-time automating of the malicious intent detection process, using a weight agnostic neural network architecture. It is an effective and accurate computational intelligent forensics tool for network traffic analysis, the demystification of malware traffic, and encrypted traffic identification in real time. Based on weight agnostic neural networks (WANNs) methodology, we propose an automated searching neural-net architecture strategy that can perform various tasks, such as identifying zero-day attacks. By automating the malicious intent detection process from the darknet, the advanced proposed solution is reducing the skills and effort barrier that prevents many organizations from effectively protecting their most critical assets. The dataset used in this study was based on CIC-

Darknet2020, which includes darknet traffic as well as corresponding normal traffic from Audio-Stream, Browsing, Chat, Email, P2P, Transfer, Video-Stream, VOIP, Files, Session and Authentication. These data are implemented either over Tor and Virtual Private Network (VPN) infrastructure or not. Details regarding the well-known cyber security dataset, their choice, and assessment can be found elsewhere [12]. Numerous publicly available real-world and simulated benchmark datasets have emerged from different sources, but their organization and adoption as standards have been inconsistent. As such, selecting and curating specific benchmarks remains an unnecessary burden. For this reason, a well-known benchmark dataset was chosen for testing our hypothesis in order to make reliable comparison experiments.

## 2. Literature Reviews

The visible layer of the web that users can access through search engines is only a small part of the internet. The part of the internet that is not accessible by search engines is also known as the Deep Web. Darknet is a subset of the Deep Web, in the sense that it is also undetectable by search engines but can be accessed with special software such as the Tor browser (see Figure 1) [13]. Tor enables users to route their traffic through "users' computers" so that traffic cannot be traced back to the originating users and conceal their identity. To pass the data from one layer to another layer, Tor has created "relays" on computers that carry information through its tunnels all over the world. The encrypted information is placed between the relays. Tor traffic as a whole passes through three relays and then it is forwarded to the final destination [14]. This mechanism ensures perfect forward secrecy between the nodes and the hidden services of Tor, while at the same time it routinely communicates through Tor nodes (consensus) operated by volunteers around the world.

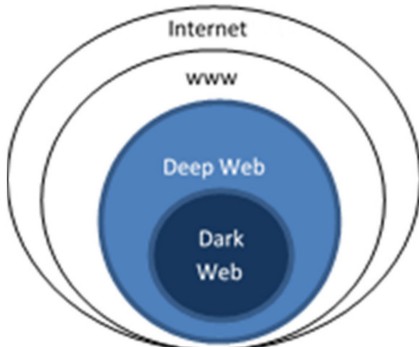

**Figure 1.** The relationship between the Internet, Deep Web and Dark Web.

Although the Tor network operates at Open Systems Interconnection (OSI) Level 4 (Transport Layer), the onion proxy software displays to clients the Socket Secure (SOCKS) interface that operates at Level 5 (Session layer). Additionally, in this network, there is a continuous redirection of requests between the retransmission nodes (entry guards, middle relays, and exit relays), with the sender and recipient addresses as well as the information being encrypted, so that no one at any point along the communication channel can directly decrypt the information or identify both ends [15].

The Tor network not only provides encryption; it is also designed to emulate the normal traffic of the Hypertext Transfer Protocol Secure (HTTPS) protocol, making the detection of Tor channels an extremely complex and specialized process, even for experienced network engineers or analysts. Specifically, the Tor network can use the Transmission Control Protocol (TCP) port 443, which is also used by HTTPS, so monitoring and identifying a session solely by the port is not a reliable method of determining this type of traffic [16].

A successful method for detecting Tor traffic involves statistically analyzing and identifying differences in the Secure Sockets Layer (SSL) protocol. SSL uses a combination

of public-key and symmetric key encryption. Each SSL connection always starts with the exchange of messages from the server and the client until a secure connection (handshake) is achieved. The handshake allows the server to prove its identity to the client using public-key encryption methods, and then allows the client and server to work together to create a symmetric key to be used to quickly encrypt and decrypt the data exchanged between them. Optionally, the handshake also allows the client to prove his identity on the server. Each Tor client generates a self-signed SSL, using a random algorithmically generated domain that changes every three minutes or so; therefore, a network traffic statistical analysis based on the specifics and characteristics of SSL can identify Tor sessions on a network combined with HTTPS traffic [8,15–17].

There is an increasing interest in research related to the dark web. A big part of the conducted literature review in cybersecurity was focused on anomaly-based network intrusion detection systems [9,17–21]. In addition, there is research dedicated to network traffic classification [22–24], whereas the Internet of Things (IoT) has recently attracted a significant amount of attention in machine learning and in network traffic analysis [13,15,16,25]. Yang et al. [26] introduce the current mainstream dark network communication system TOR and develop a visual dark web forum post association analysis system to graphically display the relationship between various forum messages and posters, which helps analysts to explore deep levels. In addition, another paper [14] designs a framework based on Hadoop in hidden threat intelligence. The framework uses a Hadoop database-based (HBase-based) distributed database to store and manage threat intelligence information, and a web crawler is used to collect data through the anonymous TOR tool in order to identify the characteristics of key dark network criminal networks, which is the basis for the later dark network research. A survey of different techniques and intrusion classification on the Knowledge Discovery in Databases KDD-Cup 99 dataset was presented by Samrin et al. [9] and an effective technique was suggested which categorized and identified intrusions in these datasets. Summerville et al. in [18], unlabeled trading data were mapped onto a set of two-dimensional grids and formed a set of bitmaps that identified anomalous and normal sessions. In the survey work of Kwon et al. [19], a review was conducted on various intrusion detection models and methodologies for classification and data volume reduction. Most of these works used the KDD-Cup 1999 dataset [20], or its successor NSL-KDD [6], which resolves some of the inherent issues of the first and has been widely adopted by the research community [17,21]. However, Zhang et al. [27] reported inefficiencies in most anomaly-based network intrusion detection systems employing supervised algorithms and suggested an unsupervised outlier detection scheme as a measure to overcome these inefficiencies. Other researchers suggested hybrid approaches for intrusion detection systems, with promising results; such as, for instance, Singh et al. [28], who combined a random forest classification technique and *k*-means clustering algorithms, and the Song et al. [29] who proposed a combination of a deep autoencoder and ensemble *k*-nearest neighbor graphs, based anomaly detectors.

Concerning network traffic classification technologies, Bayesian networks and decision tree algorithms were evaluated among others Soysal et al. in [22], and were found to suitable for traffic flow classification at high speed. Pacheco et al. in [23], a systematic review of traffic classification approaches for machine learning was made, and a set of trends is derived from the analysis performed, whereas Dhote et al. in [24], three major methods to classify different categories of Internet traffic are evaluated with their limitations and benefits. Also, a hierarchical spatial–temporal feature-based intrusion detection system (HAST-IDS) is proposed, which initially learns the low-level spatial network features of network traffic using deep convolutional neural networks (CNNs) and then learns high-level temporal features using long short-term memory networks. According to the authors, the proposed scheme demonstrates a low false alarm rate (FAR), and high accuracy and detection rate. HaddadPajouh et al. in [13], a fast and large-scale monitoring system is presented for monitoring the traffic on the darknet consisting of two parts, pre-processing and classifier. In the pre-processing part, darknet packets are transformed into a feature

vector consisting of 17 traffic features on darknet traffic. In classifier data, fast online learning is actualized by training with traffic features of known distributed denial-of-service (DDoS) attacks. The authors presented measurement results showing that the proposed solution detects backscatter packets caused by DDoS attacks with high accuracy. It also adapts very quickly to new attacks.

On the contrary, novel research has demonstrated that the assumption that the data samples collected for training machine learning models are typically assumed to be independent and identically distributed can be problematic because it simplifies the manifold of structured data. This has motivated different research areas such as data poisoning, model improvement, and the explanation of machine learning models [30]. The ability to explain, in understandable terms, why a machine learning model makes a certain prediction is becoming immensely important, because it ensures trust and transparency in the decision process of the model. Shapley values provide accurate explanations, because they assign each feature an importance value for a particular prediction [31]. For example, Messalas et al. [32] introduced a new metric, the top similarity method, which measures the similitude of two given explanations, produced by Shapley values, in order to evaluate the model-agnostic interpretability. Additionally, proposes a destructive method for optimizing the topology of neural networks based on the Shapley value, a game theoretic solution concept which estimates the contribution of each network element to the overall performance. More network elements can be simultaneously pruned, which can lead to shorter execution times and better results. An evolutionary hill climbing procedure is used to fine-tune the network after each simplification.

## 3. Methodology and Dataset

In recent years, it has been shown that advanced machine learning algorithms, such as neural networks, have the potential to be successfully applied in many areas of industry and the production process. Their success is based on the thorough processing of data that record the behavior of a system. By detecting patterns in the collected data, valuable information can be gleaned, and future predictions can be made that automate a set of processes and provide serious impetus to modern industry for value creation.

For example, multilayer neural networks, which are considered to be the easiest learning architecture, contain several linear layers that are laid out next to each other. Each of them takes an input from the previous level, multiplies it by some weights, adds a vector of bias to them, and passes the total vector through an activation function to produce the output of the level. This promotion process continues until the classification process is completed receiving the result from the final level. The final output is compared to the actual sorting values, where the sorting error is calculated using an appropriate loss function. To reduce the loss, the weights for all levels are updated one by one, using a stochastic gradient descent.

Nevertheless, their application to realistic problems remains a very complex and specialized case [33]. This is because data scientists, based on their hypotheses and experience, coordinate their numerous parameters, correlating them with the specific problems they intend to solve, utilizing the available training datasets. This is a long, tedious, and costly task.

### 3.1. MetaLearning

MetaLearning is a novel holistic approach, which automates and solves the problem of the specialized use of machine learning algorithms. It aims for the use of automatic machine learning to learn the most appropriate algorithms and hyperparameters that optimally solve a machine learning problem [34]. In particular, machine learning can be seen as a search problem, approaching an unknown underlying mapping function between input and output data. Design options, such as algorithms, model parameters (weights), hyper-parametric characteristics, and their variability, limit or expand the scope of possible mapping functions, i.e., search space.

MetaLearning techniques can discover the structures between data by allowing new tasks to be quickly learned using different types of metadata, such as the properties of the learning problem, the properties of the algorithm used (e.g., performance measures), or patterns derived from data from a previous problem. In other words, they use cognitive information from unknown examples sampled from the distribution followed by the examples in the real world, in order to enhance the result of the learning process. In this way, it is possible to learn, select, change, or combine different learning algorithms to effectively solve a given problem.

A meta-learning system should combine the following three requirements [35–38]:

1.  The system must include a learning subsystem;
2.  Experience has to be gained by utilizing the knowledge extracted from metadata related to the dataset under process or from previous learning tasks that have been completed in similar or different fields;
3.  Learning bias must be chosen dynamically.

Taking a holistic approach, a reliable meta-learning model should be trained in a variety of learning tasks and optimized for better performance in generalizing tasks, including potentially unknown cases. Each task is associated with a set of data $D$, containing attribute vectors and class tags on a supervised learning problem. The optimal parameters of the model are:

$$\theta^* = arg_\theta^{min} E_{D \sim P(D)}[L_\theta(D)] \tag{1}$$

This looks similar to a normal learning process, but a dataset is considered a sample of data.

Dataset $D$ is often divided into two parts: a training set $S$ and a set of $B$ predictions for testing and testing:

$$D = \langle S, B \rangle \tag{2}$$

$D$ datasets contain pairs of vectors and tags so that:

$$D = \{(x_i, y_i)\} \tag{3}$$

Each tag belongs to a known set of $L$ tags.

We assume a classifier $f_\theta$. The parameter $\theta$ derives the probability of a data point belonging to the class $y$ given by the attribute vector $x$, $P_\theta(y|x)$. Optimal parameters should maximize the likelihood of detecting true tags in multiple $B \subset D$ training batches:

$$\theta^* = argmax_\theta E_{(x,y) \in D}[P_\theta(y|x)] \tag{4}$$

$$\theta^* = argmax_\theta E_{B \subset D}\left[\sum_{(x,y) \in B} P_\theta(y|x)\right] \tag{5}$$

The goal is to reduce the prediction error in data samples with unknown tags, given that there is a small set of support for fast learning that works as fine-tuning.

It could be said that fast learning is a trick in which a fake dataset is created that contains a small subset of tags (to avoid exposing all the tags in the model), and various modifications are made to the optimization process in order to achieve fast learning. A brief step-by-step description of the whole process is presented below:

1.  Creation of a subset of $L_s \subset L$ tags;
2.  Creation of an $S^L \subset D$ training subset and a $B^L \subset D$ prediction set. Both of these subsets include labeled data belonging to the subset $L_s$, $y \in L_s, \forall (x,y) \in S^L, B^L$;
3.  The optimization process uses the $B^L$ subset to calculate the error and update the model parameters via error backpropagation, in the same way that it is used in a simple supervised learning model.

In this way, it can be considered that each sample pair $(S^L, B^L)$ is also a data point. Thus, the model is trained so that it can generalize to new, unknown datasets.

A modification of the supervised learning model is the following function, to which the symbols of the meta-learning process have been added:

$$\theta^* = argmax_\theta E_{L_s \subset L}\left[E_{S^L \subset D, B^L \subset D}\left[\sum_{(x,y) \in B^L} P_\theta\left(x, y, S^L\right)\right]\right] \tag{6}$$

It should be noted that retrospective neural networks with only internal memory, such as long short-term memory (LSTM), are not considered meta-learning techniques. On the contrary, the most appropriate meta-learning architecture for neural networks is neural architecture search (NAS) [39].

### 3.2. Neural Architecture Search

This is an automated learning technique for automating the design of artificial neural networks, the most widely used domain in the field of machine learning. NAS has been used to design networks that are equivalent or superior to hand-drawn architecture.

NAS methods can be categorized according to the search space, search strategy, and performance estimation strategy used [39–41]:

1. The search area determines the types of neural networks that can be designed and optimized in order to find the optimal type of neural network that can solve the given problem, e.g., a forward neural network (FFNN), recurrent neural network (RNN), etc.;
2. The search strategy determines the approach used to explore the search space, i.e., the structure of the architectural design in an internal search field of hyperparameters (levels, weights, learning rate, etc.);
3. Performance appraisal strategy evaluates the performance of a potential neural network by designing it without constructing and training it.

In many NAS methods, both micro and macro structures are searched hierarchically, allowing the exploration of different levels of standard architecture. The three NAS strategies and hierarchical search methods are shown in Figure 2.

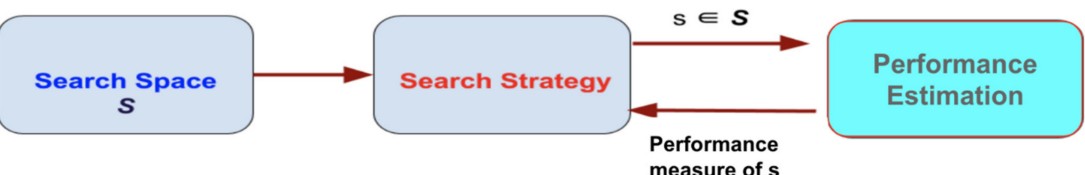

**Figure 2.** The three neural architecture search strategies.

In a hierarchical search, the first level consists of the set of primitive functions, the second level of different patterns that connect primitive functions through a directed acyclic graph, and the third level of patterns that encode how the second level patterns are connected, etc.

NAS is closely related to hyper-parameter optimization and is a subfield of automated machine learning designed to follow best practices for reducing program load, providing stable and simple environments, minimizing the number of actions required for use, providing a clear methodology for discovering knowledge in unfamiliar environments [42].

Specifically, given a neural architecture search space *F*, where the input data *D* is divided into $D_{train}$ and $D_{val}$ and the cost function Cost $(\cdot)$ (e.g., accuracy, mean squared error, etc.), the goal is to find an optimal neural network $f^* \in F$, which can achieve the lowest cost in the dataset *D*.

Finding the optimal neural network $f^*$ is equivalent to:

$$f^* = argmin_{f \in F} \, Cost(f(\theta^*), D_{val}) \tag{7}$$

$$\theta^* = argmin_\theta \ L(f(\theta), D_{train}) \tag{8}$$

where $\theta^*$ is the learning parameter of the network.

A simplified NAS procedure is described in the following Algorithm 1:

---

**Algorithm 1**

---

# **for** the number of controller epochs
**for** controller_epoch in range(controller_sampling_epochs):
    # sample a set number of architecture sequences
    sequences=sample_architecture_sequences(controller_model,samples_per_controller_epoch)
    # predict their accuracies using a hybrid controller
    pred_accuracies = get_predicted_accuracies(controller_model, sequences)
     # **for** each of these sequences
     **for** i, sequence in enumerate(sequences):
       # create and compile the model corresponding to the sequence
       model = create_architecture(sequence)
       # train said model
       history = train_architecture(model)
      # log the training metrics
      append_model_metrics(sequence, history, pred_accuracies[i])
    # use this data to train the controller
    xc, yc, val_acc_target = prepare_controller_data(sequences)
    train_controller(controller_model, xc, yc, val_acc_target)

---

The design of the NAS strategy has, as its primary objective, the definition of a neural network architecture that adapts to the nature of the dataset under consideration and to the precise coordination of ideal hyperparameters that can lead to a model with high accuracy and generalizability to data outside of training and testing sets. Typical hyperparameters that can be optimized and need to be tuned include optimization algorithms (SGD, Adam, etc.), learning rate programming, regularization, etc. [39–41]. Essentially, they enable the creation of the best learning techniques with high-performance success, with very little effort and minimal know-how.

The way the NAS strategy works can be combined with techniques based on the ways in which nature works, and in particular on finding proportions between techniques where instinct, such as a sexual characteristic, prevails over education. For example, some species in biology have predatory behaviors from the moment of their birth, which allows them to perform complex motion and sensory tasks without learning, which in many cases are completely satisfactory for the survival of the species. In contrast, in the training of artificial neurons to perform a task, an architecture that is considered suitable for modeling the task is usually chosen, and the search focuses mainly on finding the weight parameters using a learning algorithm. Inspired by social behaviors that evolved in nature, neural networks can be developed with architecture that is naturally capable of performing a given task even when weight parameters are randomized, so they can perform well without training, while their performance can be further maximized through training.

### 3.3. Proposed Method

A weight agnostic neural network (WANN) methodology [43] was used in this paper. It is an evolving strategy in neural network development techniques that can perform a specialized task regardless of the weights of the connections in the building blocks of the neural network, which equates to a lack of training [44]. The logic of using WANNs is a basic investigation in the search for architectural neural networks with specific biases that can potentially categorize the given problem, even when using random weights. By exploring such architecture, it is possible to explore factors that can perform well in their interaction environment without the need for training, which is a digital security system that can create robust self-identifying systems capable of identifying zero-day attacks.

The investigation of the WANN architectural structures used in this research fully exploits the theoretical approach of inductive bias for the production of biased results, due

to the assumptions/choices made either for the representation of the case space or for the definition of the search engine in the field of assumptions. The unrestricted space of cases is potentially infinite. By choosing the type of knowledge, i.e., the representation, this space is limited, and the first level of bias is introduced. Even so, the search space is probably still too large to perform a full search. Thus, the second level of bias is introduced, that of the search algorithm; the algorithm used does not perform a complete search in the area of possible solutions but approaches it heuristically or probabilistically. However, without these options, a learning algorithm would not be better than a random selection algorithm.

Additionally, the training data used are finite, so they do not accurately reflect the reality, because the selection process and the assumption that these data will have the same distribution as in all cases introduces another level of bias. Therefore, by reducing the inductive learning hypothesis, it is implied that the model produced by a limited set of training data will describe new, unknown cases just as well.

In conclusion, each learning algorithm has a specific bias in its elements such as representation, algorithm, or data, and this is a fundamental, necessary feature, which is taken seriously into the investigation of appropriate architectural neural networks.

Another very important process that will allow the identification of appropriate architecture to solve the given problem concerns the interpretability of the methods used. Global interpretability offers a holistic picture of the model. It is about understanding how the model makes decisions, what its most important features are, and what interactions take place between the features. A full understanding is very difficult to achieve in practice; therefore, the explanations at the universal level concern a general global representation of the model, which is not detailed and accurate. Similarly, when the explanations focus on a small area of data, then there is local interpretability, where a single example of the dataset is analyzed, and it is explained why the model made a specific decision about it. In small areas of data, the prediction may depend only linearly or monotonously on certain features, instead of having a more complex dependence on them.

Shapley values [30,31] are a very effective way of generating explanations from cooperative/coalitional game theory. The payoff/gain of the players of a cooperative game is given by a real function that gives values to sets of players. The connection of Shapley values to the problem of explaining WANN architectural structures is done in the following way. We consider the problem of WANN architectural structures as a cooperative game, whose players are the characteristics of the dataset, the profit function is the neural network model under consideration, and the model predicts the corresponding profits. In this context, the Shapley values show the contribution of each feature and therefore the explanation of why the model made a specific decision.

In conclusion, the Shapley value of characteristic $i$ of a neural network model $f$ is given by the following equation [45]:

$$\varphi_i = \sum_{S \in F \setminus \{i\}} \frac{|S|!(M - |S| - 1)!}{M!} \left[ f_{S \cup \{i\}} \left( x_{S \cup \{i\}} \right) - f_S(x_S) \right] \qquad (9)$$

where $F$ is the set of attributes, $S$ is a subset of $F$, and $M = |F|$ is the sum of the set $F$. This relationship measures the weight of each attribute by calculating its contribution when it is present in the prediction and then subtracts it when it is absent.

More specifically:

1.  $f_{S \cup \{i\}} \left( x_{S \cup \{i\}} \right)$ is the output when the $i^\infty$ attribute is present,
2.  $f_S(x_S)$ is the output when the $i^\infty$ attribute is not present;
3.  $\sum_{S \in F \setminus \{i\}} \frac{|S|!(M - |S| - 1)!}{M!}$ is the weighted mean of all possible subsets $S$ in $F$.

The SHapley Additive exPlanations (SHAP) [46] method explains model decisions using Shapley values [47]. An innovation of SHAP is that it works as a linear model, and more specifically as a method of additional contributions of features.

Intuitively with the SHAP approach, the explanation is a local linear approach to model behavior. In particular, while the model can be very complex as an integrated entity, it is easy to approach a specific presence or absence of a variable. For this reason, the degree of linear correlation of the independent and dependent variables of the set with dispersion $\sigma_X^2$ and $\sigma_Y^2$, respectively, and the covariance $\sigma_{XY} = Cov(X, Y) = E(X, Y) - E(X)E(Y)$, which is measured by calculating the Pearson's $R$ correlation table, is defined as follows:

$$R = \frac{\sigma_{XY}}{\sigma_X \sigma_Y} \qquad (10)$$

However, given the inability of the above method to detect nonlinear correlations such as sinus wave, quadratic curve, etc., or to explore the relationships between the key variables, the predictive power score (PPS) technique [48] was selected and used in this study for the predictive relationships between available data. PPS, unlike the correlation matrix, can work with non-linear relationships, with categorical data, but also with asymmetric relationships, explaining that variable A informs variable B more than variable B informs variable A. Technically, scoring is a measurement in the interval [0, 1] of the success of a model in predicting a variable target with the help of an off-sample variable prediction, which practically means that this method can increase the efficiency of finding hidden patterns in the data and the selection of appropriate forecast variables.

The use of the PPS method also focuses on the fact that a local explanation must be obtained of the models that are initially capable of operating without training and after being reinforced at a later time with training. However, the sensitivity of the SHAP method to explain the models in their hyper-parameter values, as well as the general inability to deal with the high data dimension, requires the implementation of feature selection before the application of the technique. In particular, the complexity of the problem in combination with the large number of explanations that must be given for the predictions of the model, is significantly more difficult, because the distinction between relevant and irrelevant features, as well as the distances between data points, cannot fully be captured.

Taking this observation seriously, feature selection was performed to optimally select a subset of existing features without transformation, to retain the most important of them, in order to reduce their number and at the same time retaining as much useful information as possible. This step is crucial because if features with a low resolution are selected, the resulting learning system will not perform satisfactorily, while if features that provide useful information are selected, the system will be simple and efficient. In general, the goal is to select those characteristics that lead to long distances between classes and small variations between the same class.

The process of feature selection was performed with the PPS technique, where for the calculation of PPS in numerical variables, the metric of mean absolute error (MAE) was used, which is the measurement of the error between the estimation or prediction in relation to the observed values and is calculated below [49]:

$$\text{MAE} = \frac{1}{n} \sum_{i=1}^{n} |f_i - y_i| = \frac{1}{n} \sum_{i=1}^{n} |e_i| \qquad (11)$$

where $f_i$ is the estimated value and $y_i$ is the true value. The average of the absolute value of the quotient of these values is defined as the absolute error of their relationship $|e_i| = |f_i - y_i|$.

Rescue and precision F-score (harmonic mean) was used for the categorical variables, respectively, implying that the higher the F-score, the higher the two metrics, respectively. The calculation is accomplished from the following relationship [49]:

$$\text{F}_{\text{Score}} = \frac{2 \times \text{recall} \times \text{precision}}{\text{recall} + \text{precision}} = \frac{2\text{TruePositives}}{2\text{TruePositives} + \text{FalsePositives} + \text{FalseNegatives}} \qquad (12)$$

In conclusion, to create a cybersecurity environment with fully automated solutions capable of recognizing content from the darknet, an NAS development strategy was

implemented based on the WANN technique, which was reinforced with explanations with Shapley values, having first preceded feature selection process with the PPS method.

### 3.4. Dataset

Darknet, as an overlay network, is only accessed with specific software, configurations, or licenses, often using non-standard communication protocols and ports. Its address space is not accessible for interaction with familiar web browsers, and any communication with the darknet is considered skeptical due to the passive nature of the network in managing incoming packets.

The classification of darknet traffic is very important for the categorization of real-time applications, while the analysis of this traffic helps in the timely monitoring of malware before an attack, but also in the detection of malicious activities after the outbreak.

The selection, development, or comparison of machine learning methods in novel methods can be a difficult task based on the target problem and goals of a particular study. Numerous publicly available real-world and simulated benchmark datasets have emerged from different sources, although their organization and adoption as standards have been inconsistent. As such, selecting and curating specific benchmarks remains an unnecessary burden. For this reason, we needed a well-known benchmark dataset for testing our hypothesis in order to make a reliable comparison experiment. The dataset used in this study was based on CICDarknet2020, which includes darknet traffic as well as corresponding normal traffic from Audio-Stream, Browsing, Chat, Email, P2P, Transfer, Video-Stream, VOIP, Files, Session and Authentication, which are implemented or not over Tor and VPN infrastructure. Table 1 provides details of the final categories used and the applications that implement them. Details regarding the dataset, their choice, and assessment can be found in [12,50].

**Table 1.** Darknet network traffic details.

| ID | Traffic Category | Applications Used |
|----|------------------|-------------------|
| 0 | Audio-Stream | Vimeo and YouTube |
| 1 | Audio-Stream | Crypto streaming platform |
| 2 | Browsing | Firefox and Chrome |
| 3 | Chat | ICQ, AIM, Skype, Facebook, and Hangouts |
| 4 | Email | SMTPS, POP3S and IMAPS |
| 5 | P2P | uTorrent and Transmission (BitTorrent) |
| 6 | File Transfer | Skype, SFTP, FTPS using FileZilla and an external service |
| 7 | File Transfer | Crypto transferring platform |
| 8 | Video-Stream | Vimeo and YouTube |
| 9 | Video-Stream | Crypto streaming platform |
| 10 | VOIP | Facebook, Skype, and Hangouts voice calls |

## 4. Experiments and Results

In multi-class classification, all of the indices presented below should be calculated in a one-versus-all approach. The magnitude of misclassifications is indicated by the false positive (FP) and false negative (FN) indices appearing in the confusion matrix. An FP is the number of cases where we wrongfully receive a positive result, and the FN is exactly the opposite. On the other hand, the true positive (TP) is the number of records where we correctly receive a positive result. The true negative (TN) is defined as the contrast.

The true positive rate (TPR) is also known as sensitivity; the true negative rate is also known as specificity (TNR); and the total accuracy (TA) is defined by using the below equations:

$$\text{TPR} = \frac{\text{TP}}{\text{TP} + \text{FN}} \tag{13}$$

$$\text{TNR} = \frac{\text{TN}}{\text{TN} + \text{FP}} \tag{14}$$

$$\text{TA} = \frac{\text{TP} + \text{TN}}{\text{N}} \tag{15}$$

The precision (PRE), the recall (REC), and the F-score indices are defined in the below equations:

$$\text{PRE} = \frac{\text{TP}}{\text{TP} + \text{FP}} \tag{16}$$

$$\text{REC} = \frac{\text{TP}}{\text{TP} + \text{FN}} \tag{17}$$

$$\text{F} - \text{Score} = 2 \times \frac{\text{PRE} \times \text{REC}}{\text{PRE} + \text{REC}} \tag{18}$$

In order to have a level of comparison of the proposed methodology, the dataset specified in Section 3.4 was used to identify and categorize network traffic in Tor, non-Tor, VPN, and non-VPN services in Table 1. The results of the categorization process are presented in detail for each algorithm, in Table 2. We also considered CPU time or speed (the total CPU time used by the process since it started, precise to hundredths of a second) and memory consumption as RAM hours (RAM-H) as estimates of computational resource usage. We have used a shell script based in the "top" command to monitor processes and system resource usage on the Linux OS.

**Table 2.** Classification performance metrics.

| Classifier | Accuracy | AUC | Recall | Precision | F1 | Kappa | MCC | TT (s) | RAM-H |
|---|---|---|---|---|---|---|---|---|---|
| Extreme Gradient Boosting (XGB) | 0.9012 | 0.9953 | 0.7500 | 0.9014 | 0.8990 | 0.8751 | 0.8756 | 441.61 | 0.054 |
| CatBoost | 0.8927 | 0.9942 | 0.7227 | 0.8936 | 0.8894 | 0.8642 | 0.8648 | 606.07 | 0.0662 |
| Decision Tree | 0.8858 | 0.9477 | 0.7406 | 0.8845 | 0.8849 | 0.8561 | 0.8562 | 1.90 | 0.00016 |
| Random Forest | 0.8846 | 0.9848 | 0.7245 | 0.8829 | 0.8835 | 0.8545 | 0.8545 | 19.83 | 0.00145 |
| Gradient Boosting | 0.8801 | 0.9916 | 0.7106 | 0.8797 | 0.8764 | 0.8482 | 0.8488 | 645.38 | 0.0681 |
| Extra Trees | 0.8775 | 0.9677 | 0.7201 | 0.8756 | 0.8762 | 0.8455 | 0.8455 | 11.61 | 0.00115 |
| k-Neighbors | 0.8504 | 0.9663 | 0.6748 | 0.8462 | 0.8466 | 0.8105 | 0.8108 | 7.45 | 0.00091 |
| Light Gradient Boosting Machine | 0.7826 | 0.8986 | 0.5387 | 0.7911 | 0.7808 | 0.7247 | 0.7259 | 17.41 | 0.00137 |
| Ridge Classifier | 0.6664 | 0.0000 | 0.3276 | 0.6672 | 0.6221 | 0.5659 | 0.5727 | 0.40 | 0.00006 |
| Linear Discriminant Analysis | 0.6497 | 0.9136 | 0.4400 | 0.6439 | 0.6231 | 0.5535 | 0.5575 | 2.01 | 0.00025 |
| Quadratic Discriminant Analysis | 0.3858 | 0.8710 | 0.4026 | 0.6325 | 0.4394 | 0.2936 | 0.3144 | 0.71 | 0.000087 |
| Logistic Regression | 0.3174 | 0.6756 | 0.1226 | 0.3089 | 0.2753 | 0.1237 | 0.1433 | 126.46 | 0.0135 |
| Naïve Bayes | 0.2974 | 0.6328 | 0.1281 | 0.2303 | 0.2278 | 0.0960 | 0.1120 | 0.13 | 0.00019 |
| SVM—Linear Kernel | 0.1937 | 0.0000 | 0.1037 | 0.2419 | 0.1248 | 0.0379 | 0.0485 | 130.74 | 0.0138 |
| Ada Boost | 0.1626 | 0.7142 | 0.1521 | 0.0501 | 0.0713 | 0.0788 | 0.1193 | 14.27 | 0.00129 |

AUC: Area under the Curve; MCC: Matthews Correlation Coefficient; TT: Training Time.

Additionally, the receiver operating characteristic (ROC) curves, the confusion matrix, and the class prediction error diagram of the XGBoost method, which achieved the highest success results (accuracy 90%), are presented in Figures 3–5.

In Figure 3, the colored lines in each axis represent the ROC curves. The ROC curve is a plot of the true positive rate (sensitivity) versus the false positive rate (1—specificity) as the threshold is varied. A perfect test would show points in the upper-left corner, with 100% sensitivity and 100% specificity.

Figure 4 shows the confusion matrix for testing of the darknet dataset. The XGBoost model outputs are very accurate, as determined by the high numbers of correct responses in the green squares and the low numbers of incorrect responses in the light green squares.

The class prediction error chart shown in Figure 5 provides a way to quickly understand how precise our classifier is in predicting the right classes. This plot shows the support (number of training samples) for each class in the fitted classification model as a stacked bar

chart. Each bar is segmented to show the proportion of predictions (including FN and FP) for each class. We used the class prediction error to visualize which classes our classifier had particular difficulty with, and more importantly, what incorrect answers it is giving on a per-class basis. This enables better understanding of the strengths and weaknesses of the different models and the particular challenges associated with our dataset.

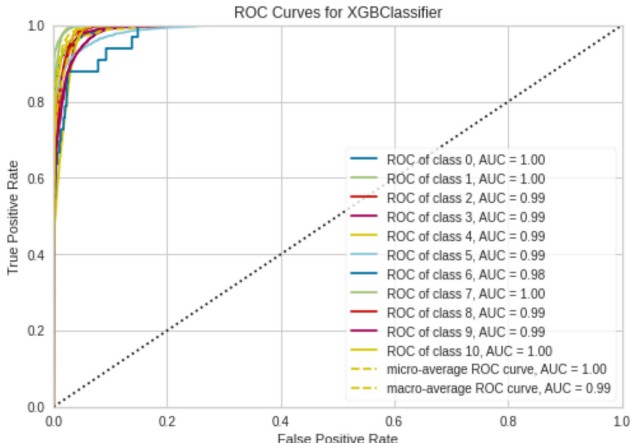

**Figure 3.** Receiver operating characteristic curves of the XGBoost classifier.

| | 0 | 1 | 2 | 3 | 4 | 5 | 6 | 7 | 8 | 9 | 10 |
|---|---|---|---|---|---|---|---|---|---|---|---|
| 0 | 361 | 0 | 65 | 1 | 0 | 2 | 0 | 11 | 0 | 5 | 0 |
| 1 | 0 | 4653 | 5 | 14 | 10 | 19 | 0 | 0 | 1 | 122 | 4 |
| 2 | 8 | 7 | 9449 | 0 | 3 | 66 | 0 | 306 | 0 | 33 | 0 |
| 3 | 2 | 30 | 40 | 2916 | 204 | 25 | 0 | 4 | 211 | 33 | 2 |
| 4 | 0 | 1 | 32 | 621 | 1054 | 9 | 0 | 1 | 116 | 3 | 0 |
| 5 | 2 | 22 | 414 | 34 | 25 | 2797 | 2 | 63 | 16 | 49 | 0 |
| 6 | 1 | 0 | 1 | 3 | 3 | 8 | 14 | 1 | 2 | 0 | 0 |
| 7 | 2 | 0 | 383 | 0 | 0 | 22 | 0 | 14153 | 0 | 5 | 0 |
| 8 | 0 | 3 | 0 | 177 | 62 | 2 | 0 | 1 | 775 | 11 | 0 |
| 9 | 8 | 399 | 300 | 6 | 9 | 59 | 0 | 20 | 6 | 2053 | 3 |
| 10 | 0 | 36 | 0 | 0 | 0 | 0 | 0 | 0 | 0 | 19 | 40 |

XGBClassifier Confusion Matrix (True Class vs Predicted Class)

**Figure 4.** Confusion matrix of the XGBoost classifier.

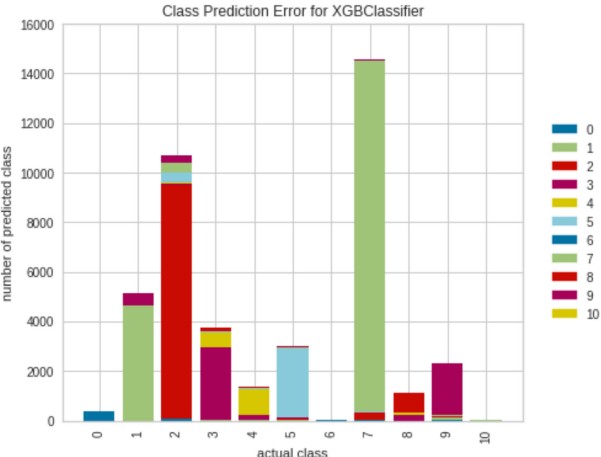

**Figure 5.** Class prediction error of the XGBoost classifier.

The automated creation of network architecture that encodes search solutions through NAS can produce architecture that, once trained, goes beyond human-designed versions. With the architecture in question, and specifically with the methodology based on the AutoKeras NAS library [51] which is designed to provide stable and simple interface environments, minimizing the number of user actions, the architecture shown in Table 3 and in Figure 6 was implemented. The results are depicted in Table 4.

**Table 3.** AutoKeras model.

| Layer (Type) | Output Shape | Parameters |
|---|---|---|
| input_1 (InputLayer) | [(None, 61)] | 0 |
| multi_category_encoding | (Mul (None, 61) | 0 |
| normalization | (Normalization (None, 61) | 123 |
| dense (Dense) | (None, 512) | 31,744 |
| re_lu (ReLU) | (None, 512) | 0 |
| dense_1 (Dense) | (None, 128) | 65,664 |
| re_lu_1 (ReLU) | (None, 128) | 0 |
| dense_2 (Dense) | (None, 11) | 1419 |
| classification_head_1 | (Softm (None, 11) | 0 |
| Total parameters: | | 98,950 |
| Trainable parameters: | | 98,827 |
| Non-trainable parameters: | | 123 |

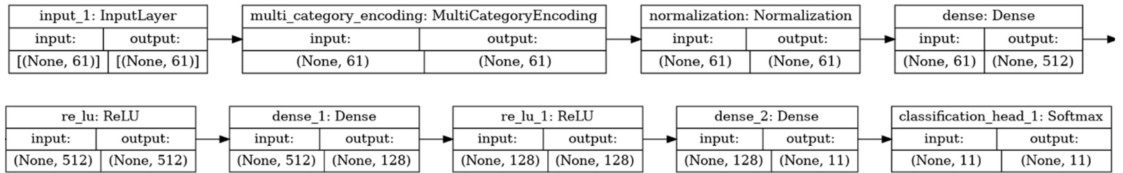

**Figure 6.** Depiction of the AutoKeras model.

**Table 4.** Classification performance metrics of the AutoKeras model.

| Classifier | Accuracy | AUC | Recall | Precision | F1 | Kappa | MCC | TT (s) | RAM-H |
|---|---|---|---|---|---|---|---|---|---|
| AutoKeras Model | 0.9268 | 0.9976 | 0.7915 | 0.9280 | 0.9105 | 0.8972 | 0.8977 | 917.23 | 0.089 |

The networks in question created by NAS, although much slower and more complex (trainable parameters: 98,827), proved to be excellent after training, as evidenced by the results of the table above.

In no case, however, can we assume that they are able to solve the given problem without training their weights. To produce architecture that satisfactorily encodes solutions, the importance of weights must be minimized. Instead of judging networks by their performance with optimal weight values, they should be evaluated by their performance when their weight values come from a random distribution. Replacing weight training with weight sampling ensures that performance is only a product of network topology and not training.

The architecture chosen to solve the given problem by a random selection of weights involves a recurring neural network with input, some sparsely connected hidden layers of reservoirs in which the choice of architecture is based on the NAS strategy, and a simple linear readout output. The connection weights in each reservoir, as well as the input weights, are random, and scaled in such a way as to ensure the echo state property, which is defined as a state where the reservoir is an "echo" of the entire input history and which is partly determined by its architecture.

The discrete levels are only those of the *u(n)* input and the *y(n)* output, while the hidden levels (reservoirs) are grouped so that the neurons are connected to each other by a

percentage that determines how sparse the network is. The synaptic compounds that unite the levels with each other are characterized by a value that determines the weights. Each input neuron is connected via $W^{in}_{ij}$ weights ($i$—input neuron; $j$—neuron to the reservoir) with weights that, although normal, are determined randomly prior to training, and their values are final because they do not change during training. Additionally, each neuron from the reservoir is connected via $W_{jk}$ weights ($j$—neuron in the reservoir; $k$—neuron in the reservoir, and $j \neq k$) to any other neuron in the reservoir. The weights of these neurons, although normal, are determined randomly before training and their values do not change. Finally, each neuron from the reservoir is connected via $W^{out}_{jm}$ weights ($j$—neuron in the reservoir; $m$—output neuron) to the output plane neurons. These weights that are in the readout layer are the only ones that are trained in order to obtain their final values [52].

The network architecture is characterized by a stacked hierarchy of reservoirs, where at each time step $t$, the first repeating layer is fed by the external input $u(t)$, while each successive layer is fed by the output of the previous one in the stack [53]. Although their architectural organization allows for general flexibility in the size of each layer, for reasons of complexity we consider a hierarchical installation of reservoirs with repeating layers $N_L$, each of which contains the same number of $N_R$ units. In addition, we use $x^{(l)}(t) \in R^{N_R}$ to declare the state of the plane $l$ at time $t$. By omitting the bias conditions, the first level state transition function is defined as follows [54]:

$$x^{(1)}(t) = \left(1 - a^{(1)}\right)x^{(1)}(t-1) + a^{(1)}tanh\ tanh\left(W_{in}u(t) + \hat{W}^{(1)}x^{(1)}(t-1)\right) \quad (19)$$

For any level greater than $l > 1$, the equation is as follows:

$$x^{(l)}(t) = \left(1 - a^{(l)}\right)x^{(l)}(t-1) + a^{(l)}tanh\ tanh\left(W^l x^{l-1}(t) + \hat{W}^{(l)}x^{(l)}(t-1)\right) \quad (20)$$

where $W_{in} \in R^{N_R \times N_U}$ is the input weight table, $\hat{W}^{(l)} \in R^{N_R \times N_R}$ is the recurrent weight table for level l, $W^{(l)} \in R^{N_R \times N_R}$ is the table relative to the connection weights between the levels from level $l$-1 to level $l$, $a^{(l)}$ is the leaky parameter at level $l$, and *tanh* represents the elementary application of the tangent [55–58].

Random weights improve the generalization properties of the solution of a linear system because they produce almost rectangular (weakly correlated) features. The output of a linear system is always correlated with the input data; therefore, if the range of solution weights is limited, rectangular inputs provide a wider range of solutions than those supported by weights. Additionally, small weight fluctuations allow the system to become more stable and noise resistant, because input errors will not be amplified at the output of a linear system with little correlation between input and output weights. Thus, the random classification of weights that produces weakly correlated characteristics at the latent level allows achieving a satisfactory solution and a good generalization performance.

Essentially, for random forward propagation architecture with a hidden plane and random representation of hidden plane neurons, the input data ARE mapped to a random *L*-dimensional space with a distinct set of training N, where $(x_i, t_i)$, $i \in [\![1, N]\!]$ με $x_i \in R^d$ και $t_i \in R^c$. The network output is represented as follows [56,59]:

$$f_L(x) = \sum_{i=1}^{L} \beta_i h_i(x) = h(x)\beta\ i \in [\![1, N]\!] \quad (21)$$

where $\beta = [\beta_1, \ldots, \beta_L]^T$ is the output of the weight table between the hidden nodes and the output nodes, $h(x) = [g_1(x), \ldots, g_L(x)]$ are the outputs of the hidden nodes (random hidden attributes) for input $x$, and $g_1(x)$ is the exit of the $i$ hidden node. The basis of an $N$ set of training $\{(x_i, t_i)\}_{i=1}^{N}$, can solve the learning problem $H\beta = T$, where $T = [t_1, \ldots, t_N]^T$ the target labels and the output table of the hidden level $H$ as below:

$$H(\omega_j, b_j, x_i) = \left[g(\omega_1 x_1 + b_1) \cdots g(\omega_l x_1 + b_l) \vdots \ddots \vdots g(\omega_1 x_N + b_1) \cdots g(\omega_l x_N + b_l)\right]_{N \times l} \quad (22)$$

Prior to training, the input weight table $\omega$ and the bias vectors $b$ are randomly generated in the interval $[-1, 1]$, with $\omega_j = [\omega_{j1}, \omega_{j2}, \ldots, \omega_{jm}]^T$ and $\beta_j = [\beta_{j1}, \beta_{j2}, \ldots, \beta_{jm}]^T$. The output level table of the hidden level $H$ is calculated from the activation function and the use of the training data based on the following function:

$$H = g(\omega x + b) \tag{23}$$

The output weights $\beta$ can be calculated from the relationship:

$$\beta = \left( \frac{I}{C} + H^T H \right)^{-1} H^T X \tag{24}$$

where $H = [h_1, \ldots, h_N]$ are the outputs of the hidden level and are the input data. $\beta$ can be calculated from the generalized inverse Moore-Penrose table:

$$\beta = H^+ T \tag{25}$$

where $H^+$ is the generalized inverse Moore–Penrose table for table $H$.

In this case, the proposed standardization offers the possibility of managing multiple intermediate representations, because the hierarchical organization of random reservoirs architecture in successive layers naturally reflects the structure of the dynamics of the developed system. This scaling allows the progressive classification and exploration of input data interfaces across the levels of the hierarchical architecture, even if all levels share the same weight values. Furthermore, the multilevel architecture represents a transitional state of how the internal representations of the input signals are determined, which guarantees high performance even for problems that require long internal memory intervals. It also has higher performance in cases where short-term network memory capabilities are required than the corresponding architecture, which would have to work with the same total number of iterative or retrospective units in order to achieve corresponding results. In addition, in terms of computational efficiency, the multilevel construction of reservoirs in the design of a neural system also results in a reduction in the number of non-zero repetitive connections, typically associated with other types of retrospective architecture. This implies low complexity and time savings required to perform specialized tasks.

These conclusions are reflected in Table 5 below, which shows the very high categorization results (accuracy 94%) in combination with the very short processing time (290 s, which is 66% faster than the corresponding AutoKeras model):

**Table 5.** Classification performance metrics of the reservoir model.

| Classifier | Accuracy | AUC | Recall | Precision | F1 | Kappa | MCC | TT (s) | RAM-H |
|---|---|---|---|---|---|---|---|---|---|
| Reservoir Model (13-11-09) | 0.9451 | 0.9988 | 0.8122 | 0.9317 | 0.9242 | 0.9108 | 0.9094 | 290.08 | 0.0338 |

However, even in this case there was weight training, and therefore the result of the process is a product of training. Unlike WANN, weight training is avoided. Focusing exclusively on exploring solutions in the field of neural network topologies, using random common weights for each network level and recording the cumulative result during the test, reservoir architecture was used, without weight training.

To identify the number of resulting solutions, the process was assisted with explanations using the Shapley value methods, after first selecting features with the PPS method. The resulting network population was then ranked according to their performance and complexity so that the highest-ranking networks were selected to form a new solution population. The process was repeated until the best architecture was found. The architecture was modified either by inserting a node by separating an existing connection, by adding a connection by connecting two previously unconnected nodes, and by changing the activation function which reassigns activation functions.

Initially, the predictive power of the problem variables was analyzed to identify the variables with the highest PPS, in order to identify the most important ones that can solve the problem, simplifying the process, and at the same time without reducing the effectiveness of the method. From the total of variables, 19 were selected with a significant score greater than 0.3, while the rest had a predictive capacity of less than 0.1.

A summary of the 19-variable PPS capture table is presented in Table 6.

**Table 6.** Predictive power score.

| Idle_Max | Idle_Mean | Idle_Min | Packet_Length_Max |
|---|---|---|---|
| 0.471 | 0.444 | 0.430 | 0.399 |
| Packet_Length_Mean | Average_Packet_Size | Flow_IAT_Max | Fwd_IAT_Max |
| 0.383 | 0.379 | 0.372 | 0.363 |
| Bwd_Packet_Length_Max | Fwd_Packet_Length_Max | Total_Length_of_Bwd_Packet | Bwd_Packet_Length_Mean |
| 0.349 | 0.345 | 0.340 | 0.338 |
| Bwd_Segment_Size_Avg | Total_Length_of_Fwd_Packet | Packet_Length_Std | Packet_Length_Variance |
| 0.338 | 0.338 | 0.330 | 0.330 |
| Fwd_Header_Length | Subflow_Bwd_Bytes | Fwd_Packet_Length_Mean | - |
| 0.328 | 0.320 | 0.313 | |

Extensive research was then conducted on evaluating the values of the variables, how they contribute to the prediction, and explaining each decision of the implemented models, using the Shapley values. Figure 7 shows the classification of the values of the variables used in the bar plot.

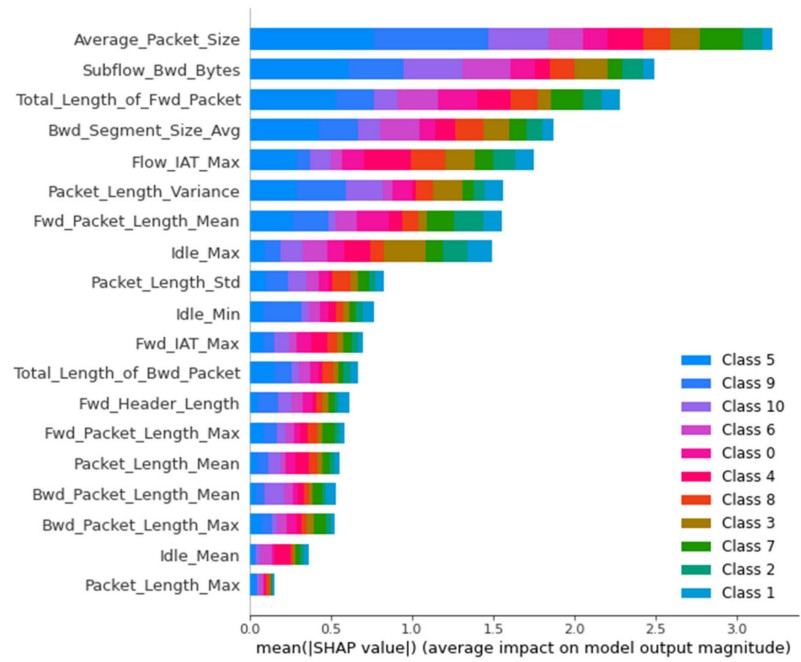

**Figure 7.** SHapley Additive exPlanations (SHAP) value impact on model output (bar plot).

In Figure 8 is presented the summary beeswarm plot, which is the best way to capture the relative effect of all the features in the whole dataset. Characteristics are classified based on the sum of Shapley values in all samples in the set. The most important features of the model are shown from top to bottom. Each attribute consists of dots, which symbolize each attribute of the package, while the color of the dot symbolizes the value of the attribute (blue corresponds to a low value, while red corresponds to a high value). The position of the dot on the horizontal axis depends on its Shapley value.

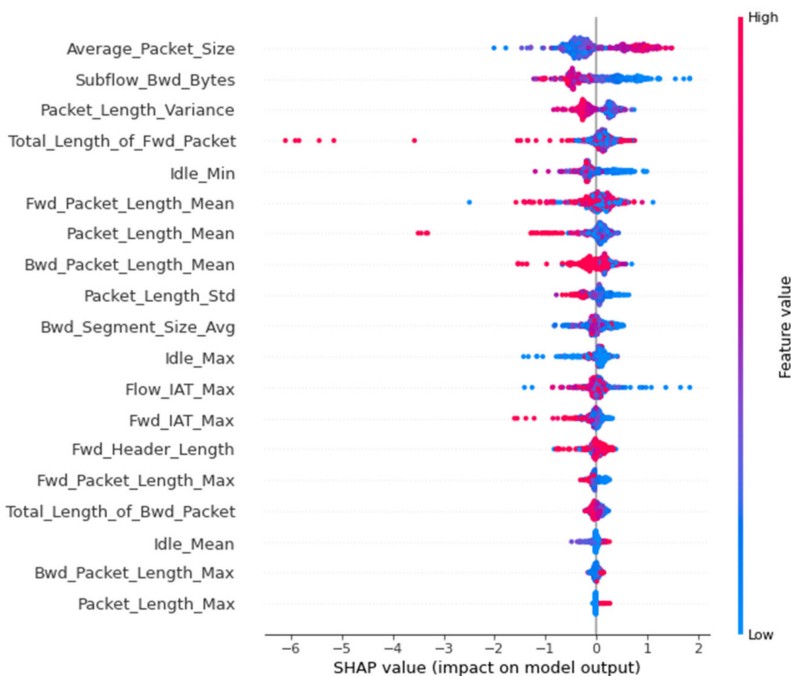

**Figure 8.** SHAP value impact on model output.

We see that the Average_Packet_Size attribute is the most influential for the model predictions. Additionally, for its high values (red dots), the Shapley value is also high, so it has a great positive effect, i.e., it increases the probability that the package under consideration comes from the darknet. On the contrary, for its low values (blue dots), the Shapley value is low, so it has a negative effect on the forecast, i.e., it increases the probability that the package under consideration does not come from darknet.

In Figure 9, a sample selection is used from the dataset to represent the typical attribute values, and then 10 samples are used to estimate the Shapley values for a given prediction. This task requires $10 \times 1 = 10$ evaluations of the model.

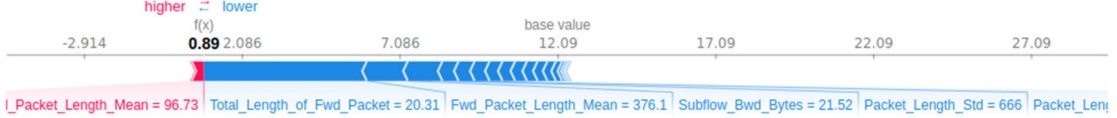

**Figure 9.** Explanation of a single prediction (10 evaluations).

In this diagram, a local explanation is presented, where the base value refers to the average value of the model forecasts; in this case, the model predicts that the data package analyzed comes from the darknet with a probability of 12%. For this package, the forecast value is 89%, so the Shapley values show the change from the average forecast to the specific forecast. The red arrows push the prediction to the right, i.e., they help to increase the probability that this package comes from darknet, while the blue arrows push to the left, helping to reduce the probability that it comes from darknet. The length of each arrow symbolizes the magnitude of the effect on the prediction. In this example, we see that the Fwd_Packet_Length_Mean attribute helps to increase the likelihood that the package will come from the darknet (Shapley value 96.73), while the Total_Length_of_Fwd_Packet (Shapley value 20.31) and Fwd_Packet_Length_Mean (Shapley value 376.1), decrease this likelihood, etc.

Shapley values also have universal explanation capabilities, summing the values of a set of samples. In the image below are used a selection of 100 samples from the dataset

to represent the standard attribute values, and then 500 samples are used to estimate the Shapley values for a given prediction. This task requires $500 \times 100 = 50,000$ model evaluations. Figure 10 below represents the explanation of a single prediction of 50,000 evaluations.

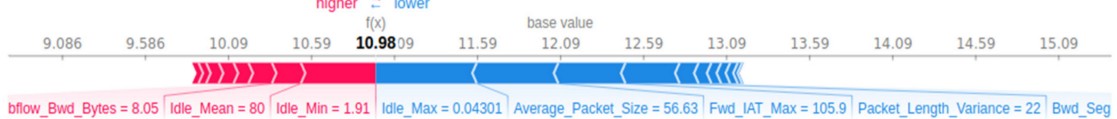

**Figure 10.** Explanation of a single prediction (50,000 evaluations).

In Figure 11, there is a diagram of the above process of 50,000 validations, but with an explanation of multiple predictions and their fixation in relation to their similarity.

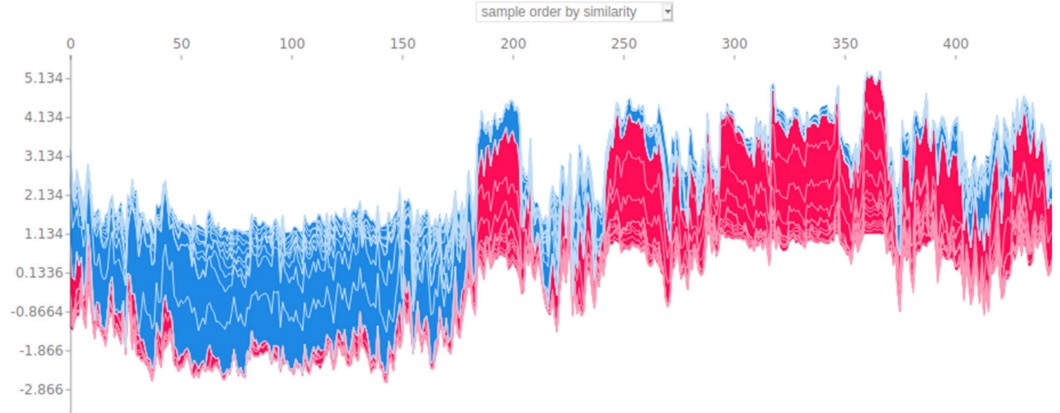

**Figure 11.** Explanation of many predictions by similarity.

In Figure 12, the same procedure is captured based on the output values of the model.

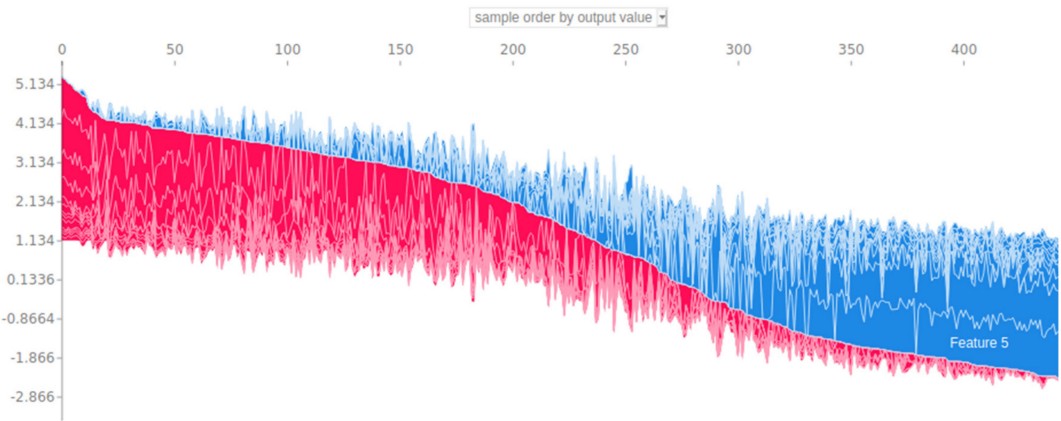

**Figure 12.** Explanation of many predictions by output value.

In both cases, sampling is used to implement the explanation. It should be noted that for the evaluation of the architecture that was finally selected, thorough and long-term research was carried out on the effects of the features used in each prediction, taking into account local and universal explanations from the Shapley values methodology.

The resulting architecture is presented in the Table 7, with the corresponding categorization performance metrics. The values in parentheses indicate the size and depth of the reservoirs; for instance, the Reservoir model (11-17-09) that performed with the highest

results had a depth of 3, i.e., it included 3 reservoirs, which incorporated 11, 17, and 09 neurons, respectively.

**Table 7.** Classification performance metrics of the proposed Reservoir models.

| Classifier | Accuracy | AUC | Recall | Precision | F1 | Kappa | MCC | TT (s) | RAM-H |
|---|---|---|---|---|---|---|---|---|---|
| Reservoir (11-17-09) | 0.8392 | 0.9904 | 0.7502 | 0.8910 | 0.9004 | 0.8973 | 0.8807 | 59.35 | 0.00676 |
| Reservoir (06-11-05-12) | 0.7961 | 0.9892 | 0.7483 | 0.8884 | 0.8798 | 0.8657 | 0.8623 | 68.16 | 0.00843 |
| Reservoir (15-10-06-08) | 0.7504 | 0.9879 | 0.7469 | 0.8534 | 0.8419 | 0.8479 | 0.8588 | 53.22 | 0.00663 |
| Reservoir (21-18-13) | 0.7492 | 0.9809 | 0.7450 | 0.8511 | 0.8415 | 0.8468 | 0.8501 | 72.38 | 0.00897 |
| Reservoir (05-12-16-07) | 0.7262 | 0.9654 | 0.7203 | 0.8397 | 0.8402 | 0.8414 | 0.8428 | 64.04 | 0.00781 |

By attempting an evaluation of the above results, it is easy to conclude that the proposed framework is a particularly remarkable learning system that in all evaluation cases achieved remarkable results in relation to the respective competing systems, always taking into account that the proposed system competes with corresponding systems which received training, while the competing ones did not.

The characteristic of this is that the Reservoir model (11-17-09) that gave the highest results surpassed the algorithms Light Gradient Boosting Machine, Ridge Classifier, Linear Discriminant Analysis, Quadratic Discriminant Analysis, Logistic Regression, Naïve Bayes, SVM—Linear Kernel, and Ada Boost. This observation can be interpreted in the non-linearity that generally characterizes neural networks, especially in the case where it is examined where the mechanism of production of input signals is non-linear. The resistance to structural errors of neural networks, and especially the sparse architecture of reservoirs, guarantees that the malfunction or destruction of a neuron or some connections is not able to significantly disrupt their function because the information they contain is not located in a specific point but diffuse throughout the network, and especially in cases where the Echo State Property (ESP) property that characterizes reservoir architecture is achieved.

Another important observation is that the method produces extremely accurate results without recurring problems of an undetermined cause because all the features in the dataset in question are handled very efficiently, based on the original feature selection processes performed on the basis of PPS. This resulted in the removal of insignificant features in the network, which under certain conditions can be characterized as noise, with very negative effects on the final result.

In addition, one of the main advantages gained from the results is the high reliability resulting from kappa prices (high reliability if $k \geq 0.70$) [49], which can be considered as the result of data processing that allows the retention of the most relevant data for the forthcoming forecasts.

Finally, the use of the reservoir technique in this work is related to the fact that very often in multifactorial problems of high complexity, such as the one under consideration, the prediction results are multivariate, which can be attributed to the sensitivity of the correlational models in the data. The two most important advantages of this technique focus on the fact that it offers better predictability and stability, because the overall behavior of the model is less noisy while the overall risk of a particularly poor choice that may result from modeling is reduced. The above view is also supported by the dispersion of the expected error, which is concentrated close to the average error value, a fact that categorically states the reliability of the system and the generalization ability that it presents.

An important weakness of the methodology followed is the highly specialized and time-consuming preprocessing procedure that must be followed to identify the appropriate architecture that can perform satisfactorily, which adds research complexity to the data analysis and explanation models used.

## 5. Discussion and Conclusions

In this paper, an innovative, reliable, low-demand, and highly efficient network traffic analysis system was presented, which relies on advanced computing intelligence

methods. The proposed framework implements and utilizes the advantages of meta-learning methodology, in order to identify malfunctions or deviations of the normal mode of operation of the network traffic, which, in most cases, is the outcome of cyber-attacks. The proposed digital security system was tested on a complex dataset that responded to specialized operating scenarios of normal and malicious network behavior. Our motivation in this work was to explore the processes to which only neural network architecture, without prior learning, can codify solutions and model a given task. The holistic approach proposed, which automates and solves the problem of specialized use of neural network finding algorithms without the need for human experience or intervention, is a promising approach to capture an unknown underlying mapping function between input and output data in a given problem, without requiring system training.

Under this consideration, this paper proposes an innovative and highly effective weight agnostic neural network framework for darknet traffic, big-data analysis, and network management, to real-time automate the malicious intent detection process. The proposed architecture, which is first implemented and presented in the literature, facilitates more specialized pattern recognition systems without prior training, which are capable of responding to changing environments. It is important to emphasize that the proposed method removes complexity from the way NAS strategies work, because it utilizes multiple functions of specialized methods for extracting useful intermediate representations in complex neural network architecture. The initial utilization of the predictive power of the independent variables significantly reduces the computing, producing improved training stability and remarkable categorization accuracy. In addition, the reservoir technology used leads for remarkable forecasting results, implementing a robust forecasting model capable of responding to highly complex problems. This ability was found in the high convergence speed of the proposed architecture, which was calculated by a simple array calculation, in contrast to reciprocating stochastic gradient descent models. To prove the validity of the proposed methodology, SHAP methodology was used, which is based on the evaluation of solutions based on Shapley values. This technique provided an understanding of how the model makes decisions and what interactions take place between the features used. Additionally, for the precise design of the specific search space of the best architectural prototypes that can optimally solve the problem, the relationships between the variables were explored and feature selection was implemented with the predictive power score technique, in order to briefly measure the predictive relationships between the available data.

In conclusion, the paper presents a method for the development of interpretable neural networks by encoding the solution directly in the network architecture and not in the training of its weights. Compared to other learning methods, the proposed architecture is resistant in changes to node inputs, which could be the foundation for a robust defense against adversarial attacks or even damaged networks.

Proposals for the development and future improvements of this framework should focus on the automated optimization of the appropriate parameters of method pre-training to achieve an even more efficient, accurate, and faster categorization process. It would also be important to study the expansion of this system by implementing more complex architecture with Siamese neural networks in an environment of parallel and distributed systems [60] or over blockchain [61,62]. Finally, an additional element that could be studied in the direction of future expansion concerns the operation of a network with methods of self-improvement and re-defining its parameters, so that the process of selecting the architectural hyper-parameters can be fully automated in order to identify dark web services [63,64], exploits from the dark web [65–68], and classify malicious traffic from this network [69–71].

**Author Contributions:** Conceptualization, K.T., D.T., K.D., L.I. and C.S.; methodology, K.T., D.T., K.D. and C.S.; valida-tion, K.T., D.T., K.D., L.I. and C.S.; formal analysis, K.T., D.T., K.D. and C.S.; investigation, K.T., D.T., K.D. and C.S.; writing—original draft preparation, K.T., D.T., K.D, L.I. and C.S.; writing—review and editing, K.T., D.T., K.D., L.I. and C.S.; supervision, C.S.; project administration, K.D., L.I. All authors have read and agreed to the published version of the manuscript.

**Funding:** This research received no external funding.

**Conflicts of Interest:** The authors declare no conflict of interest.

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
