# Peer review of "Darknet Traffic Big-Data Analysis and Network Management for Real-Time Automating of the Malicious Intent Detection Process by a Weight Agnostic Neural Networks Framework"

_electronics, doi:10.3390/electronics10070781_

Round 1

Reviewer 1 Report

This work proposes a network management and analysis framework and is evaluated with darknet traffic. It is based on weight agnostic neural networks (WANNs).

The first half of the journal is unnecessarily lengthy (up to page 12), as most of the content is existing literature and could have been just summarised along with relevant references.

Poor presentation and description of the actual work. Sections 1-3, are mainly focused on describing the current issues for network traffic analysis, and providing general comments about malware and Darknet. The actual work proposed in this ms is not introduced until the end of Section 3, noticeably within a section titled “Tor hidden services on Darknet”.

Literature review briefly mention disparate topics, without going into critically analysing the literature, and describing how these are relevant for the proposed work.

Section 5 describes in extent meta-learning and NSA. Subsection 5.3 Proposed Method, continues providing generic comments more adequate for the literature review section. Poor description of the dataset in Section 5, which not contribute to understand the relevance of this data in the empirical results.

Section 6 presents results without context; ROC and bar charts lack correlation between the graphs and the approaches used to generate the results; badly described empirical approach.

Academic writing could be improved.

Author Response

Dear respected reviewer ,

We would like to thank you for your prompt response reply to our paper entitled "Darknet Traffic Big-Data Analysis and Network Management to Real-Time Automating the Malicious Intent Detection Process by a Weigh Agnostic Neural Networks Framework". We would also like to thank you for their constructive remarks and suggestions, which we believe we have fully addressed by making the necessary revisions that have resulted in substantial improvement of the overall quality of our manuscript.

……………………….

Yours sincerely,

Konstantinos Demertzis, Konstantinos Tsiknas, Dimitrios Taketzis, Charalabos Skianis and Lazaros Iliadis

Comment 1. The first half of the journal is unnecessarily lengthy (up to page 12), as most of the content is existing literature and could have been just summarised along with relevant references.

Answer. Thanks for the comment, we have substantially reduced the contents of the first half of the manuscript. In particular, we have completely removed Section 2 (Network Management) and 3 (Tor hidden services on Darknet) and summarised their contents in Section 1 (Introduction), and in Section 3 (Literature Reviews) (please see pages from 3 to 9)

Comment 2. Poor presentation and description of the actual work. Sections 1-3, are mainly focused on describing the current issues for network traffic analysis, and providing general comments about malware and Darknet. The actual work proposed in this ms is not introduced until the end of Section 3, noticeably within a section titled “Tor hidden services on Darknet”.

Answer. We have now described the actual work in the introduction, please see page 3 last paragraph: In this paper, …

For this reason, a well-known benchmark dataset was chosen for testing our hypothesis in order to make reliable comparison experiments.

Comment 3. Literature review briefly mention disparate topics, without going into critically analysing the literature, and describing how these are relevant for the proposed work.

Answer. The comparison against other Machine Learning Algorithms is performed in Section 3 (Metalearning), please see for instance below:

For example, multilayer neural networks, which are considered to be the easiest learning architectures, contain several linear layers that are laid out next to each other…   

Nevertheless, their application to realistic problems remains a very complex and specialized case. This is because data scientists, based on their hypotheses and experience, coordinate their numerous parameters, correlating them with the specific problems they intend to solve, utilizing the available training data sets. This is a long, tedious and costly task.

we also perform a comparative analysis in the last five paragraphs of section 4:  

The characteristic of this is that the Reservoir model (11-17-09) that gave the highest results ....An important weakness  of the methodology followed is the highly specialized and time-consuming preprocessing process that must be followed to identify the appropriate architectures that can perform satisfactorily, which adds research complexity to data analysis and explanation models used.

Comment 4. Section 5 describes in extent meta-learning and NSA. Subsection 5.3 Proposed Method, continues providing generic comments more adequate for the literature review section. Poor description of the dataset in Section 5, which not contribute to understand the relevance of this data in the empirical results.

Answer 4. Thank you for this constructive comment. The Section 5 reduced by half according to the reviewer’s comments and suggestions. Also, the selection, development, or comparison of machine learning methods in novel methods is a complex task based on the target problem and the goals of the particular study. Numerous publicly available real-world and simulated benchmark datasets have emerged from different sources, but their organization and adoption as standards have been inconsistent. As such, selecting and curating specific benchmarks remains an unnecessary burden. For this reason, a well-known benchmark dataset was chosen for testing our hypothesis in order to make reliable comparison experiments. The data set used in this study is based on CICDarknet2020, which includes Darknet traffic as well as corresponding normal traffic from Audio-Stream, Browsing, Chat, Email, P2P, Transfer, Video-Stream, VOIP, Files, Session and Authentication, which are implemented or not over Tor and VPN infrastructures. Details regarding the well-known cyber security dataset, their choice and assessment can be found in Reference [12]. We have updated the introduction accordingly (please see page 3 last paragraph).

Comment 5. Section 6 presents results without context; ROC and bar charts lack correlation between the graphs and the approaches used to generate the results; badly described empirical approach.

Answer 5. We would like to thank the reviewer for this constructive comment that gives us the opportunity to clarify things further. We have added detailed explanations in the Experiments and Results section. Specifically, in Multi-class classification all of the indices presented below should be calculated in an one versus all approach. The magnitude of misclassifications is indicated by the False Positive (FP) and False Negative (FN) indices appearing in the confusion Matrix. A FP is the number of cases where we wrongfully receive a positive result and the FN is exactly the opposite. On the other hand, the True Positive (TP) is the number of records where we correctly receive a Positive result. The True Negative (TN) is defined respectively. The True Positive rate (TPR) also known as Sensitivity, the True Negative rate also known as Specificity (TNR), the Total Accuracy (TA, the Precision (PRE) the Recall (REC) and the F-Score indices are defined by using the appropriate equations in the manuscript. We also added in the Results tables the CPU Time or Speed (the total CPU time used by the process since it started, precise to the hundredths of a second) and memory consumption as RAM Hours (RAM-H) as estimates of computational resources usage. We have used a shell script based on the "top" command to monitor processes and system resource usage on the Linux OS. In addition, we have added an explanation about ROC curves. Specifically, the ROC curve is a plot of the true positive rate (sensitivity) versus the false positive rate (1 - specificity) as the threshold is varied. A perfect test would show points in the upper-left corner, with 100% sensitivity and 100% specificity. Finally, we have added detailed explanations about the confusion matrix and for the Class Prediction Error chart that provides a way to quickly understand how good our classifier is at predicting the right classes. This plot shows the support (number of training samples) for each class in the fitted classification model as a stacked bar chart. Each bar is segmented to show the proportion of predictions (including FN and FP) for each class. We used the Class Prediction Error to visualize which classes our classifier is having a particularly difficult time with, and more importantly, what incorrect answers it is giving on a per-class basis. This enables us to better understand the strengths and weaknesses of different models and particular challenges unique to our dataset.

Comment 6. Academic writing could be improved

Answer 6. Thank you for this constructive comment. We have rearranged the entire paper, have corrected the typos and grammar errors and have adapted the text to the Academic writing required by a scientific journal. 

Reviewer 2 Report

The paper proposes a traffic analysis platform with the intention of identifying malicious activity by means of a weight agnostic neural network.

Note I am not an expert in the topic of the paper.

I believe the article is well written and presents novel research although some sections are unnecessarily long and can be reduced in size, for instance Section 4 and Section 5 (7.5 pages). Section 5 includes details of concepts that can be easily summarized as well as referenced with a citation. Again, for the sake of helping the reader spotting out the main results of the paper I would suggest the reduction of Conclusion Section. References are well up to date, although I would recommend reducing their number (74 are a lot). I would also recommend the Authors to include some indication of the incurred overhead (resources, computational time,...) of using such platform in production.

There are also minor issues such as English spelling typos, so I would recommend a revision regarding this aspect.

Author Response

Dear respected reviewer,

We would like to thank you for your prompt response reply to our paper entitled "Darknet Traffic Big-Data Analysis and Network Management to Real-Time Automating the Malicious Intent Detection Process by a Weigh Agnostic Neural Networks Framework". We would also like to thank you for their constructive remarks and suggestions, which we believe we have fully addressed by making the necessary revisions that have resulted in substantial improvement of the overall quality of our manuscript.

……………………….

Yours sincerely,

Konstantinos Demertzis, Konstantinos Tsiknas, Dimitrios Taketzis, Charalabos Skianis and Lazaros Iliadis

Comment 1. I believe the article is well written and presents novel research although some sections are unnecessarily long and can be reduced in size, for instance Section 4 and Section 5 (7.5 pages). Section 5 includes details of concepts that can be easily summarized as well as referenced with a citation.

Answer 1. Thank you for the remarks and for the careful reading. We have substantially reduced the contents of the first half of the manuscript. In particular, we have completely removed Section 2 (Network Management) and 3 (Tor hidden services on Darknet) and summarised their contents in Section 1 (Introduction), and in Section 3 (Literature Reviews). Also, the Sections 4 and 5 reduced by half.

Comment 2. Again, for the sake of helping the reader spotting out the main results of the paper I would suggest the reduction of Conclusion Section.

Answer 2. The Conclusion section reduced by half according to the reviewer’s comments and suggestions. Thank you for this helpful comment.

Comment 3. References are well up to date, although I would recommend reducing their number (74 are a lot).

Answer 3. Thank you for this constructive comment. We have rearranged the entire paper and reducing significant the references (62) according to the reviewer’s comments and suggestions.

Comment 4. I would also recommend the Authors to include some indication of the incurred overhead (resources, computational time,...) of using such platform in production.

Answer 4. We would like to thank the reviewer for this constructive comment that gives us the chance to clarify things further. The results of the categorization process are presented in detail for each algorithm, in Table 2. We also considered CPU Time or Speed (the total CPU time used by the process since it started, precise to the hundredths of a second) and memory consumption as RAM Hours (RAM-H) as estimates of computational resources usage. We have used a shell script based in the "top" command to monitor processes and system resource usage on the Linux OS.

Comment 5. There are also minor issues such as English spelling typos, so I would recommend a revision regarding this aspect.

Answer 5. Thank you for this constructive comment. We have rearranged the entire paper, have corrected the typos and grammar errors and have adapted the text to the Academic writing required by a scientific journal.